# Enhanced electron dephasing in three-dimensional topological insulators

Jian Liao[1], Yunbo Ou[2], Haiwen Liu[3], Ke He[2], Xucun Ma[2], Qi-Kun Xue[2] & Yongqing Li[1,4,5]

Study of the dephasing in electronic systems is not only important for probing the nature of their ground states, but also crucial to harnessing the quantum coherence for information processing. In contrast to well-studied conventional metals and semiconductors, it remains unclear which mechanism is mainly responsible for electron dephasing in three-dimensional topological insulators (TIs). Here, we report on using weak antilocalization effect to measure the dephasing rates in highly tunable $(Bi,Sb)_2Te_3$ thin films. As the transport is varied from a bulk-conducting regime to surface-dominant transport, the dephasing rate is observed to evolve from a linear temperature dependence to a sublinear power-law dependence. Although the former is consistent with the Nyquist electron-electron interactions commonly seen in ordinary 2D systems, the latter leads to enhanced electron dephasing at low temperatures and is attributed to the coupling between the surface states and the localized charge puddles in the bulk of 3D TIs.

[1] Beijing National Laboratory for Condensed Matter Physics, Institute of Physics, Chinese Academy of Sciences, Beijing 100190, China. [2] State Key Laboratory of Low Dimensional Quantum Physics, Department of Physics, Tsinghua University, Beijing 100084, China. [3] Center for Advanced Quantum Studies, Department of Physics, Beijing Normal University, Beijing 100875, China. [4] School of Physical Sciences, University of Chinese Academy of Sciences, Beijing 100190, China. [5] Beijing Key Laboratory for Nanomaterials and Nanodevices, Beijing 100190, China. Correspondence and requests for materials should be addressed to K.H. (email: kehe@mail.tsinghua.edu.cn) or to Y.L. (email: yqli@iphy.ac.cn).

Three-dimensional topological insulators (TIs) have emerged as an important class of materials that are characterized by an insulator-like bulk and gapless surface states protected by time-reversal symmetry[1,2]. TIs and their derivative structures have been predicted to possess many fascinating properties and attracted a great deal of attention[1–4]. Recent progresses in improving the quality and electrical gating of TI materials[5–11] have led to remarkable observations of the quantum anomalous Hall effect[12], the quantum Hall effect[13,14], as well as many quantum coherent transport properties, such as weak antilocalization (WAL)[15–17], Aharonov-Bohm and Aharonov-Aronov-Spivak effects[18,19] and universal conductance fluctuations[20,21]. It has also been proposed that quantum interference experiments can be used to probe Majorana zero modes formed on the TI surfaces due to superconducting proximity effect[22,23]. A full understanding of the dephasing processes is thus important for utilising various phase-coherent properties in TIs[4,24]. Such insight is also crucial to addressing many fundamental questions regarding TI surface states, for instance, the nature of their ground states under the influence of disorder and electron–electron interactions[25–28]. However, the electron dephasing rates obtained in experiments exhibit a variety of temperature-dependent behaviours. Some groups reported the linear temperature dependences[19,20,29,30] that are often encountered in conventional 2D electron systems[31–33], whereas others found much weaker or stronger temperature dependences even in TI samples with presumably insulating bulk[34–36]. The lack of consistency in the temperature dependences has precluded clear identification of the dephasing mechanisms, and brings an obstacle to the quantum coherent experiments that require long electron dephasing lengths[4,22–24].

Measurement of the magnetoresistance due to weak localization or WAL has proven a reliable technique to study the electron dephasing in diffusive electron systems of various dimensions[31–33]. Similar magnetotransport measurements have been carried out on many types of TI thin films or microflakes[10,11,15–17,29,30,34–38]. The low field magneto-conductivity is usually described very well with the Hikami-Larkin-Nagaoka (HLN) equation[39] at the strong spin-orbit coupling limit:

$$\Delta\sigma(B) = \sigma_{xx}(B) - \sigma_{xx}(0)$$
$$= -\alpha\frac{e^2}{\pi h}\left[\psi\left(\frac{1}{2} + \frac{B_\phi}{B}\right) - \ln\left(\frac{B_\phi}{B}\right)\right], \quad (1)$$

where $\psi(x)$ is the digamma function, $B_\phi = \frac{\hbar}{4De\tau_\phi} = \frac{\hbar}{4el_\phi^2}$ is the dephasing field, $D$ is the diffusion constant, and $l_\phi$ is the dephasing length. For a single-channel WAL-type transport,

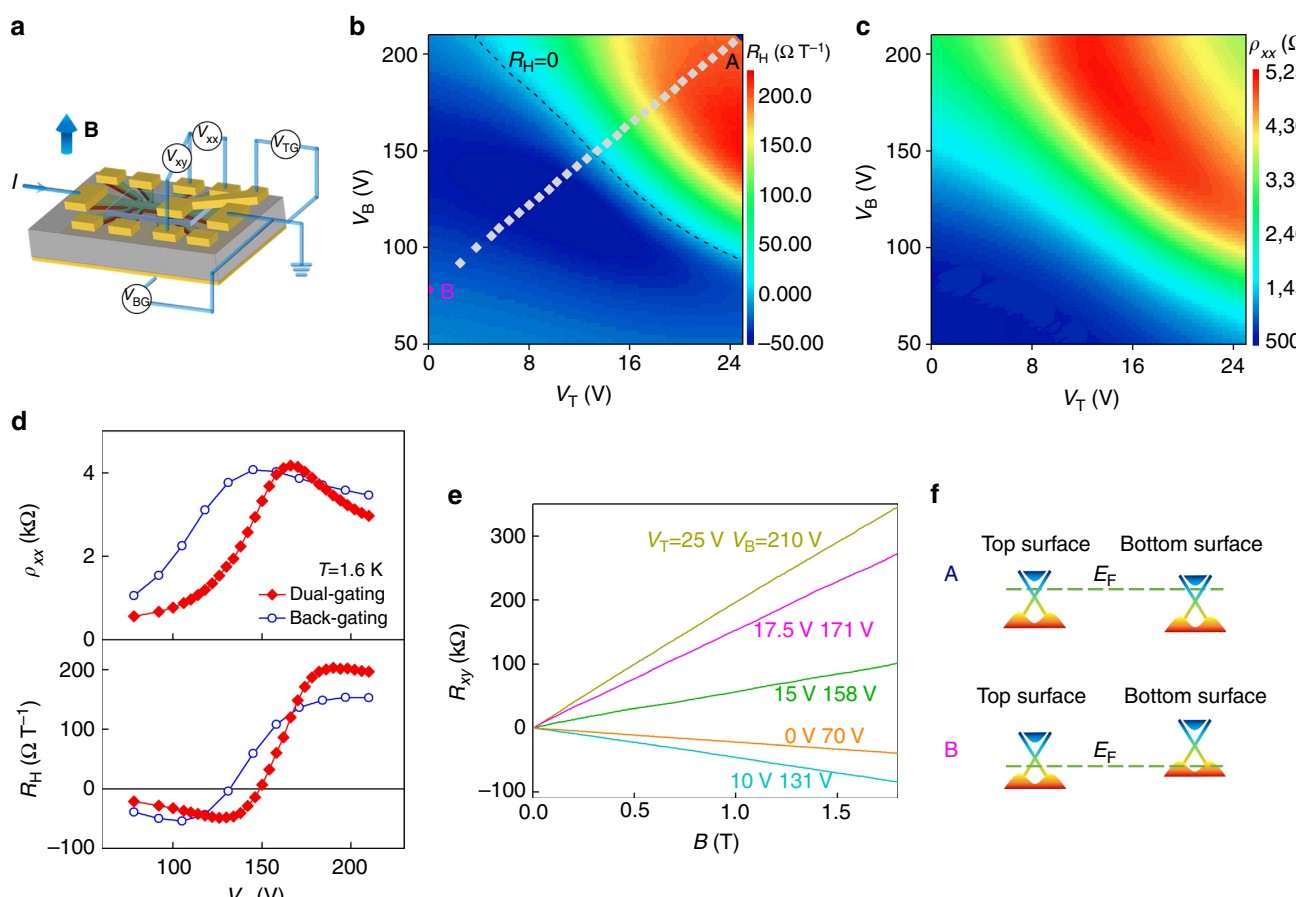

**Figure 1 | Tunable electron transport in a 15 nm thick (Bi$_{1-x}$Sb$_x$)$_2$Te$_3$ topological insulator (TI) thin film (Sample #1).** (**a**) Sketch of the Hall bar-shaped device with a back gate and a top gate. The width of the current path is 50 μm. (**b,c**) Hall coefficient $R_H$ and longitudinal sheet resistivity $\rho_{xx}$ rendered in a 2D map in top-gate voltage $V_T$ and bottom-gate voltage $V_B$. The thin dashed line in **b** marks the gate voltages for $R_H = 0$. The sign of $R_H$ is set to positive for electrons throughout this work. (**d**) Gate-voltage dependence of zero-field $\rho_{xx}$ and Hall coefficient $R_H$. The open circles represent the data for $V_T = 0$, whereas the solid symbols stand for the case of dual-gating, in which the corresponding ($V_T$, $V_B$) values are shown in **b** with solid diamonds. (**e**) Hall resistance curves for several set of top and bottom gate voltages. Data in **b**–**e** were taken at $T = 1.6$ K. (**f**) Schematic band diagrams for the top and bottom surface states for the decoupled surface-transport regime (A) and the bulk-conducting regime (B).

prefactor $\alpha$ would be equal to 1/2. In case of multi-channel WAL, the $\alpha$ value can vary from 1/2 to $n_c/2$, where $n_c = 2\alpha$ is the number of parallel conduction channels. The $n_c$ value obtained from the HLN fit is, however, often far from integers due to the difference in the dephasing fields or the coherent coupling between these channels[40]. When the inter-channel coupling is not negligible, determination of the dephasing rate $\tau_\phi^{-1}$ becomes very challenging because the parameter $B_\phi$ extracted from the HLN fit is no longer a simple quantity proportional to $\tau_\phi^{-1}$. An ideal scenario is the surface-dominant transport with two symmetric channels (corresponding to $\alpha = 1$, see Supplementary Note 1). It allows for straightforward extraction of the dephasing rate with a fit to Equation (1). This transport regime, however, requires not only the bulk is insulating, but also the top and bottom surfaces are decoupled and have identical dephasing fields. Unfortunately, most of the dephasing measurements reported so far have not fulfilled these conditions owing to inadequate control of the surface and bulk conductivities.

In this work, we present the measurements of electron dephasing rates in 3D TI thin films with highly tunable chemical potential. The phase coherent transport related to the WAL can be tuned continuously from a bulk-conducting regime with $\alpha = 1/2$ to a decoupled surface-transport regime with $\alpha = 1$ in a single device. Whereas the common Nyquist dephasing behaviour[41] is observed in the former regime, the dephasing rate is found to have a sublinear power-law temperature dependence in the surface-transport regime. We propose that the coupling between the surface states and localized bulk states in a variable range hopping (VRH) regime is responsible for the enhanced electron dephasing and the sublinear temperature dependence in the surface-transport regime.

## Results

**Characterization of $(Bi_{1-x}Sb_x)_2Te_3$ field-effect devices**. Our measurements were carried out in a set of field-effect devices based on 15–30 nm thick $(Bi_{1-x}Sb_x)_2Te_3$ (BST) films grown on $SrTiO_3$ substrates with molecular beam epitaxy. A high Sb composition ($x > 0.9$) was chosen to prevent the Dirac point being buried inside the bulk valence band[8,9]. The ungated BST films have a conducting bulk with $p$-type carriers. As illustrated in Fig. 1 with a 15 nm thick BST film (Sample #1), the Hall-bar shaped device is equipped with both the top and bottom gates, which enable a large range tuning of the chemical potential. When a positive gate voltage is applied, the hole density is reduced and correspondingly the magnitude of Hall coefficient $R_H$ decreases. When the gate voltage is sufficiently high, the Fermi level passes the charge neutral point, which is manifested as a reversal of the sign of $R_H$ and appearance of a maximum in longitudinal resistivity $\rho_{xx}$. At large positive gate voltages, the Fermi level is shifted into the bulk band gap and the surface-dominant transport is achieved. For instance, with back-gate voltage $V_B = 210$ V and top-gate voltage $V_T = 25$ V, surface carrier densities as low as $n_1 = 3.3 \times 10^{11}$ cm$^{-2}$ and $n_2 = 4.0 \times 10^{12}$ cm$^{-2}$ can be obtained from a two-band fit of the Hall effect data. Such low electron densities indicate that the Fermi level in the BST film resides in the bulk band gap (See Supplementary Fig. 2 and Supplementary Note 2 for more details), consistent with angular resolved photoemission studies of similar BST films[9].

**High tunable phase-coherent transport**. Figure 2a shows the magnetoconductivity curves of the BST film at several gate voltages at $T = 1.6$ K. All of them can be satisfactorily fitted with the HLN equation. Shown in Fig. 2b are the $\alpha$ and $B_\phi$ values extracted from the fits. For small gate voltages, $\alpha$ is close to 1/2, in

agreement with previous measurements of TI thin films with conducting bulk[10,11,15,16,37]. This can be attributed to strong surface-bulk coupling, which makes the sample, behaving like a single-channel system in the phase coherent transport despite the coexistence of multiple conduction channels[40]. This transport regime is realized when the inter-channel scattering rate is much higher than the dephasing rates in individual channels (See Supplementary Note 1). It is also noteworthy that in this work the dephasing length $l_\phi = (D\tau_\phi)^{1/2} = \left(\frac{\hbar}{4eB_\phi}\right)^{1/2}$ extracted from the fit to Equation (1) is always much longer than the corresponding film thickness. Therefore, even in the bulk-conducting regime, the phase coherent transport is two dimensional, justifying the application of the HLN equation for the data analysis. As the positive gate voltage increases, the depletion of hole carriers in the bulk leads to gradual decoupling between the top and bottom surfaces and accordingly the $\alpha$ value becomes greater[15–17,38]. When these two surfaces are fully decoupled and their dephasing fields are brought into equality, $\alpha = 1$ is observed. For the BST thin films studied in this work, both $\alpha \simeq 1/2$ and $\alpha \simeq 1$ can be maintained for a wide range of temperatures at fixed gate voltages, as depicted in Fig. 2d,e. Such a good tunability in the phase coherent transport provides a solid foundation for studying the temperature dependence of dephasing rate in TIs.

**Electron dephasing rates in surface-transport regime**. Figure 3a shows the temperature dependence of the dephasing field $B_\phi$ extracted from the HLN fits for the case of $\alpha \simeq 1$. In this decoupled surface-transport regime, the transport takes place in two independent, equivalent channels and the dephasing rate $\tau_\phi^{-1}$ is simply proportional to $B_\phi$ (See Supplementary Note 1). Figure 3b depicts that $B_\phi$ has a sublinear power-law dependence: $B_\phi \propto T^p$ with $p = 0.55$ for a wide range of temperatures ($\sim 0.1$–10 K). Similar results have also been obtained from other BST samples with various thicknesses (15–30 nm), in which $p$ is distributed in a range of 0.45–0.60 when the prefactor $\alpha$ is tuned close to 1 (See Supplementary Fig. 6, Supplementary Table 1 and Supplementary Note 5). These values are substantially lower than the $p = 1$ that is corresponding to the linear dependence commonly observed in conventional 2D electron systems[31,32].

**Crossover to linear dependence in bulk-conducting regime**. In contrast to the surface-transport regime, the dephasing field in the bulk-conducting regime has qualitatively different temperature dependences. As depicted in Fig. 4a, $B_\phi$ exhibits a nearly perfect linear temperature dependence when $\alpha$ is tuned close to 1/2. Figure 4b,c (see Supplementary Fig. 4 for more details) further show that reducing the bulk conductivity by increasing the gate voltage can induce a crossover from the linear $T$-dependence to sublinear ones by varying the gate voltage. The variation of $p$ from $\sim 1$ to 0.55 is correlated well with the increase of $\alpha$ from $\sim 1/2$ to $\sim 1$, as the bulk-mediated coupling between the top and bottom surfaces become stronger with increasing bulk conductivity. Observation of such a crossover in a single device rules out the possibility of extrinsic mechanisms, such as magnetic impurities[4], environmental microwave radiation[42] or interfacial interaction with the STO substrates, for the sublinear $T$-dependences in the decoupled surface-transport regime. Otherwise, similar deviations from the linear dependence would also be observed in the bulk-conducting regime.

## Discussion

For a weakly disordered conventional 3D electron system, electron–phonon interactions are the dominant source of electron dephasing, which gives rise to a power-law temperature dependence of the dephasing rate: $\tau_\phi^{-1} \sim T^p$, with $p = 2$–4 (ref. 32). In low

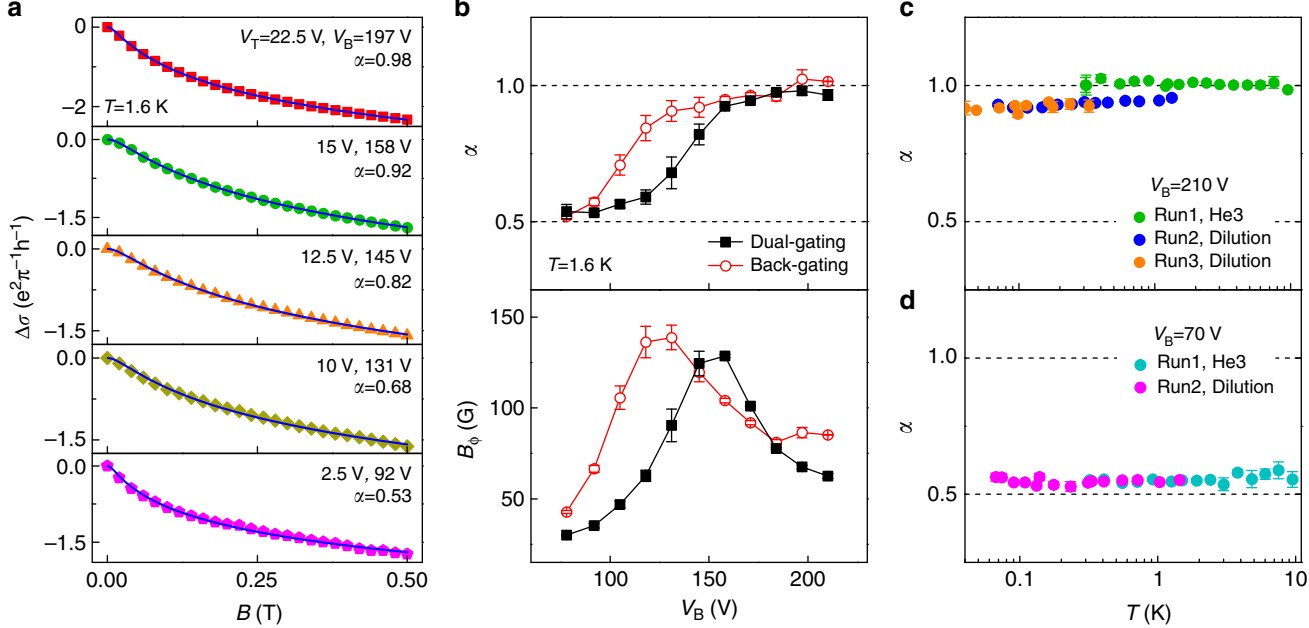

**Figure 2 | Tunable surface-bulk coupling in the BST thin film (Sample #1) revealed in the magnetotransport.** (**a**) Low field magnetoconductivity and the best fits to the Hikami-Larkin-Nagaoka (HLN) equation. (**b**) Gate-voltage dependence of prefactor $\alpha$ and dephasing field $B_\phi$ extracted from the HLN fits. The gate voltages used for the dual-gating are same as those in Fig. 1. The error bars denote s.d.s of $B_\phi$ determined from various fitting ranges. (**c,d**) Temperature dependence of the prefactor $\alpha$ in both decoupled surface-transport regime ($\alpha \approx 1$) and bulk-conducting regime ($\alpha \approx 1/2$). Slight differences between different cool-downs were caused by exposure of the sample to atmosphere during the measurement intervals.

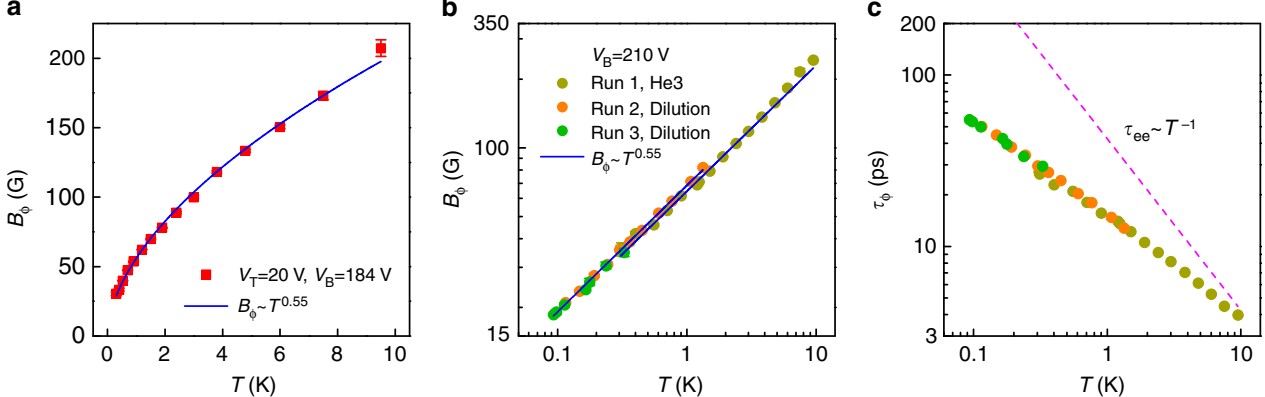

**Figure 3 | Enhanced electron dephasing in the decoupled surface-transport regime.** (**a**) Temperature dependence of dephasing field $B_\phi$ for a dual-gating case with $\alpha \approx 1$. The data can be well fitted to $B_\phi \propto T^p$ with $p = 0.55$. (**b**) $T$-dependences of $B_\phi$ and the corresponding power-law fits for three cool-downs with single-gating. The extracted exponent $p$ is same as the dual-gating case. (**c**) Dephasing time $\tau_\phi$ of the bottom surface states as a function of temperature. The dashed line shows the estimated dephasing times due to the Nyquist electron–electron interactions.

dimensional systems, the dominant dephasing mechanism at low temperatures is usually associated with small energy transfer processes owing to electron–electron interactions[31]. This so-called Nyquist dephasing, first proposed by Altshuler et al.[41], also leads to a power-law $T$-dependence, but with a smaller exponent: $p = 2/3$ and 1 for 1D and 2D electron systems, respectively. The Nyquist mechanism has been confirmed by numerous magnetotransport experiments on low dimensional metals or semiconductors[31,32]. As shown above, the $p$ values for the decoupled surface-transport regime in TIs are in a range of 0.45–0.60, substantially lower than those of the known dephasing mechanisms for weakly disordered, nonmagnetic 2D electron systems[31–33]. Figure 3c further shows that dephasing times estimated for the surface states are considerably shorter than the theoretical values for the Nyquist dephasing. This indicates the existence of an additional dephasing source in the topological transport regime.

Given the fact that decreasing the bulk conductivity in TIs can induce the crossover from the Nyquist dephasing to the sublinear power-law temperature dependence, it is reasonable that the evolution of bulk states with gating are involved in the crossover of the dephasing behaviour. As revealed by previous studies, TIs in the family of bismuth chalcogenides do not have truly insulating bulk[3]. Even in the state-of-the-art materials, such as $(Bi,Sb)_2(Te,Se)_3$ single crystals[5–7,13], the bulk conductivity does not freeze out completely at low helium temperatures (typically on the order of $\Omega \cdot cm$). This was first explained by Skinner et al.[43], who pointed out that the narrow band gaps and electrostatic fluctuations from the compensation doping result in the formation of localized nanometre-sized charge puddles in the bulk, and consequently the VRH of charge carriers between these puddles is energetically favoured over the thermal activation. In addition to the estimated bulk resistivities from

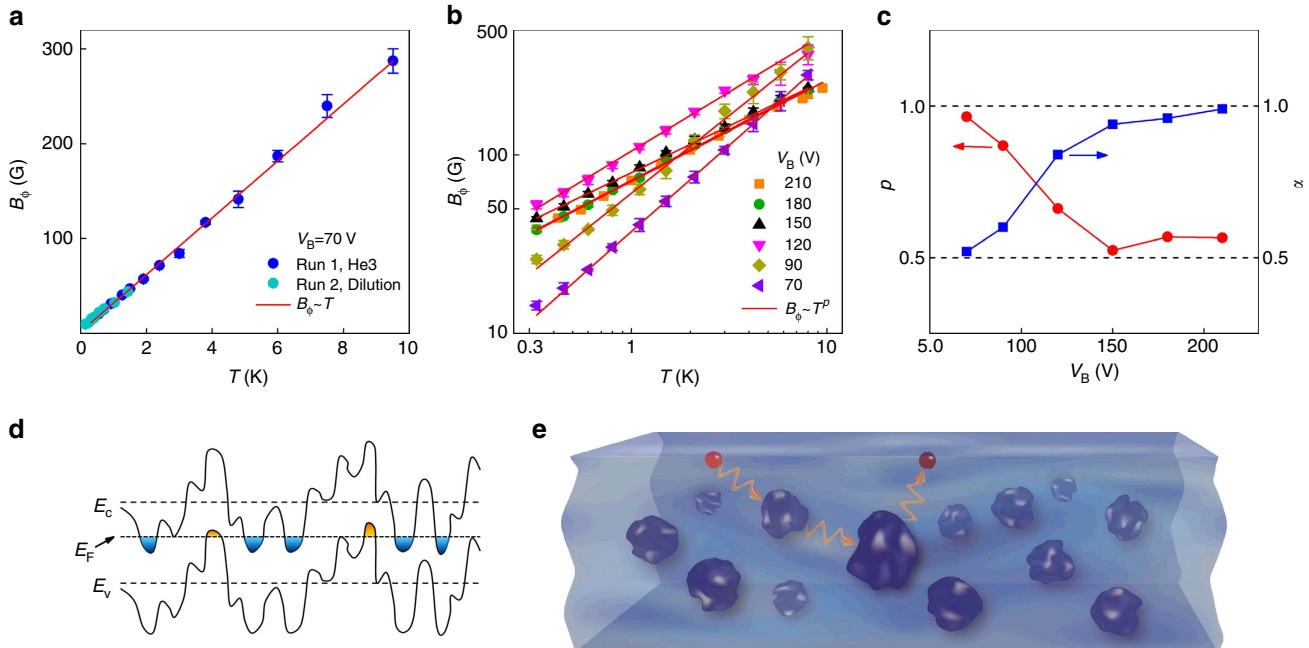

**Figure 4 | Tunable power-law temperature dependence of the dephasing rate.** (**a**) Temperature dependence of dephasing field $B_\phi$, which is proportional to the dephasing rate $\tau_\phi^{-1}$, in the bulk-conducting regime (symbols) and the corresponding linear fit (line). (**b**) $T$-dependences of $B_\phi$ for several gate voltages. The transport is tuned from the bulk-conducting regime ($\alpha \approx 1/2$) to the decoupled surface-transport regime ($\alpha \approx 1$) as the gate voltage $V_B$ increases. The $B_\phi$ values obtained from the HLN fits are shown in symbols and the lines are the fits to $B_\phi \propto T^p$. (**c**) Prefactor $\alpha$ and exponent $p$ for the same set of gate voltages as (**b**). (**d**) Schematic diagram of the TI bulk bands with strong fluctuations due to the compensation doping. The filled areas denote nanoscale electron and hole puddles[43]. (**e**) Cartoon illustration of electron dephasing in the surface states owing to strong coupling to the localized charge puddles in the bulk.

the transport measurements of thick TI single crystals, the existence of surface and bulk charge puddles has been supported by scanning tunnelling microscopy[44] and optical conductivity measurements[45], respectively. In addition, the VRH transport has been directly observed in ultrathin BST films, in which the surface conductivity is suppressed by a hybridization gap[46].

For the BST films with the bulk layer in the VRH regime, the transport is dominated by the diffusive Dirac fermions on the surfaces. If the films are sufficiently thick, the top and bottom surfaces do not couple to each other coherently owing to the lack of direct tunnelling. The phase coherent transport can be modelled as a decoupled two-channel system. Even though the localized charge puddles carry little electrical current due to the high resistivity, they can couple to the surface states via tunnelling. As the hopping transport is an inelastic process, the surface-bulk coupling makes the charge puddles working as an environmental bath with very low energy excitations and thus opens a new avenue for the dephasing in surface states. In the VRH regime, the dephasing length is believed to be set by the hopping distance and hence follows $l_{\phi,\text{VRH}} \propto T^{-p/2}$, with $p = 2/3$ and $1/2$ for 2D and 3D systems, respectively[47] (See Supplementary Note 3). At sufficiently low temperatures, $l_{\phi,\text{VRH}}$ becomes shorter than that for the Nyquist dephasing, namely $l_{\phi,\text{ee}} \propto T^{-p/2}$ with $p = 1$. Therefore, the surface-bulk coupling can reduce the dephasing length significantly, leading to the enhanced dephasing rates shown in Fig. 3c. In case of $l_{\phi,\text{VRH}} \ll l_{\phi,\text{ee}}$ and strong surface-bulk coupling (that is, $\tau_{\phi,\text{SB}} \ll \tau_{\phi,\text{ee}}$), the dephasing rate follows $\tau_\phi^{-1} \propto l_{\phi,\text{VRH}}^{-2} \propto T^p$ with $1/2 \leq p \leq 2/3$. This is consistent with the $p$ values ($p = 0.45$–$0.60$) obtained from the dephasing field measurements as well as the temperature dependence of conductivity (Fig. 4, Supplementary Figs 3–6 and Supplementary Notes 4–6).

In addition to the dephasing induced by the hopping between the localized bulk states, the inelastic coupling between the surface states and individual charge puddles might also cause dephasing. For 2D TIs, Väyrynen et al.[48,49] showed theoretically that electron tunnelling between edge states and individual charge puddles in the bulk can cause electron backscatterings and hence a decrease in conductivity. An earlier theoretical work by Jiang et al.[50] showed that similar deviation from the quantized conductance in 2D TIs can be understood as an electron dephasing effect. However, a direct link between the dephasing and the charge puddles in 2D TIs has been established neither in theory nor in experiment[51]. To the best of knowledge, previously reported works on charge puddles in 3D TIs were focused on the conductivity of the bulk[43–45], and the possible influence on electron dephasing has never been investigated. Even though it is reasonable that the strong coupling between the charge puddles and the edge/surface states could exist in both 2D and 3D TIs, there could be some qualitative differences between these two systems. It was pointed out in ref. 50 that only the dephasing scatterings with spin flips can cause electron backscatterings in 2D TIs. In contrast, all types of electron dephasing, including those without spin flips, are expected to cause a suppression of the WAL effect in 3D TIs. It is also interesting to note that a recent theoretical work showed that the energy transport in the VRH regime is much more efficient than the charge transport due to Coulomb interaction between the localized states[52]. It is likely that the Coulomb interaction of the TI surface states with the charge puddles in the bulk can potentially be an additional source of dephasing. Obviously, further theoretical efforts are needed to gain deeper insight into the electron dephasing or more generally the quantum transport properties related to the charge puddles in both 2D and 3D TIs.

As all known TIs have narrow band gaps in the bulk[3], the electron dephasing caused by the localized charge puddles should not be limited to the BST thin films studied in this work. At low temperatures, the enhanced dephasing severely shortens the dephasing length, and should be detrimental to the experiments requiring long phase coherent lengths, for instance, the interferometers proposed to detect Majorana zero modes[22–24]. Therefore, it is highly desirable to search for the TI materials in which the dephasing caused by the charge puddles can be substantially suppressed. The electron dephasing measurement will play an unreplaceable role in such endeavours because it can probe the interaction between the surface and bulk states at a very small energy scale (down to the order of μeV). Such information is very difficult, if not impossible, to obtain from other transport or spectroscopic measurements. Finally, it is noteworthy that the coupling between the diffusive surface states and the bulk states in the hopping regime revealed in this work can be extended to non-topological bilayer systems. Such coupling can be utilized to offer a new method to measure electron dephasing rates in electron systems that are not in the weakly disordered regime. This could provide valuable information to the study of electron dephasing phenomena in non-diffusive systems, which have been very challenging both experimentally and theoretically[53].

## Methods

**Thin film growth and device fabrication.** TI BST thin films were grown on 500 μm thick $SrTiO_3$ (111) substrates in a molecular beam epitaxy (MBE) system with a base pressure of $1 \times 10^{-10}$ Torr or lower. The BST films are single crystalline and have large, atomically flat terraces on the surfaces, similar to those reported previously[46,54]. They were capped with a 10 nm thick amorphous Te layer before being taken out of the molecular beam epitaxy chamber. The samples were subsequently patterned into standard Hall bars with photolithography and chemical wet etching. A 35 nm thick $AlO_x$ thin film was deposited with atomic layer deposition onto the BST samples to serve as the dielectric material for top-gating. The $SrTiO_3$ substrates, which have exceptionally large dielectric constants and high electrical breakdown strength, were used as the bottom-gate dielectric. Both top and bottom gate-electrodes, as well as electrical contacts were prepared by thermal evaporation of Cr/Au thin films with typical thicknesses of 5 nm/80 nm.

**Electron transport measurements.** The $I$–$V$ characteristics of all electrical contacts and gates had been checked with Agilent source-measurement units with a current resolution of 100 fA before the electron transport measurements. All of the data presented in this manuscript were taken from the samples with good ohmic contacts and the leakage current is negligible for both top and bottom gates. The transport experiments were performed in a vapour-flow $^4$He flow cryostat, a $^3$He refrigerator and a top loading $^3$He/$^4$He dilution refrigerator with magnetic fields up to 15 T. Electrical wirings of the dilution refrigerator are equipped with a variety of low pass filters at different temperature stages so that electron temperatures lower than 15 mK can be obtained. The electron temperature has been confirmed by the measurements of activation gaps of fragile fractional quantum Hall states (for instance, the 5/2 state), and dephasing phenomena in the quantum Hall plateau-to-plateau transitions in high mobility GaAs/AlGaAs 2D electron systems, as well as quantum transport properties of other systems. Standard lock-in technique with low frequency ac current (typically at 17.3 Hz and no more than 100 nA) was used for the transport measurements. The amplitude of excitation current was optimized to avoid electron heating effects while maintaining sufficient signal-to-noise ratios (see Supplementary Fig. 1 for an example of testing the measurement current). The longitudinal and Hall resistivities presented in this manuscript were obtained by symmetrization and anti-symmetrization of the raw magnetotransport data with respect to the magnetic fields.

**Data availability.** The data that support the findings of this study are available from the corresponding author upon request.

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

## Acknowledgements

We thank P. Ostrovsky, Q. F. Sun and X. C. Xie for valuable discussions. This work was supported by the National Basic Research Program of China (Projects No. 2015CB921102, No. 2015CB921001 and No. 2012CB921703), the National Key Research and Development Program (Project No. 2016YFA0300600), the National Science Foundation of China (Projects No. 61425015, No. 11374337, No. 11325421, No. 11674028 and No. 91121003), and the Strategic Initiative Program of Chinese Academy of Sciences (Project No. XDB070200).

## Author contributions

Y.L. and K.H. initiated the project. Y.O. carried out the MBE growth of the samples. J.L. fabricated the Hall bar devices, performed the electron transport measurements. J.L., H.L. and Y.L. analysed the data and prepared the manuscript with contributions from all authors.

## Additional information

**Competing interests:** The authors declare no competing financial interests.

