## [Peer Review File · Nature Communications]

Reviewers' comments:

Reviewer #1 (Remarks to the Author):

In the manuscript, the authors studied the dephasing mechanism in topological insulators (TIs). By tuning the Fermi level of TIs with top and back gates, the bulk became insulating. In such decoupled surface transport regime, a sublinear power law dependence of dephasing rate is observed. The authors ascribed it to the existence of charge puddles in the bulk of TIs due to the small band gap and the compensation doping. This study not only reveals a possible new dephasing mechanism in TIs, but also gains insight into the ground state of TIs. It is important to the community of TIs and suitable for publication in Nature Communications. But if such bulk-state charge puddles do exist in the bulk of TIs, they are also expected to appear at the surface, as well as the surface-state puddles with different charge types. Can such bulk-state and surface-state puddles at the surface also lead to the dephasing of carriers? Can we observe any experimental phenomena related with such charge puddles in the decoupled surface transport regime?

Reviewer #2 (Remarks to the Author):

Liao and co-workers report on enhanced electron dephasing in the three-dimensional topological insulator $(\text{Bi,Sb})_2\text{Te}_3$. The authors tune their transport properties using the gating technique to probe different regimes in this material. In their comprehensive experimental study, the authors claim that they observe a change in the dephasing rate and attribute their observation to surface transport or coupling between surface states and localized charge puddles in the bulk.

As mentioned by the authors in the introductory part, literature is full of electron dephasing experiments on topological insulators and I do not see that the present paper has the novelty to be of interest for a broad physics community and therefore, I do not recommend publication of this paper in Nature Communications. This work could be interesting for a very limited community working on quantum interference experiments on novel materials and I think that Physical Review B might be an appropriate journal for publication.

I have the following comments that should be addressed by the authors:

The authors claim that their dephasing experiments can disentangle different transport regimes in TIs, however, to my opinion, this assumption is not justified.

- 1) I do not see experimental evidence for surface-dominant transport. The authors use a two-carrier model and claim that the Fermi level is in the bulk band gap. What is the evidence for 2D transport? Can they exclude that a low-concentration bulk band instead of a surface state contributes to transport?
- 2) How should I understand the statement 'the surfaces are fully decoupled' or 'gradual decoupling between top and bottom surface'? Do the authors mean hybridization? Is there any experimental evidence for that?
- 3) How does the total concentration depend on the applied gate voltage(s)?
- 4) Can the authors comment on the presence of trivial surface states and possible geometric effects that can be present in thin TI films?

Before attributing the different regimes to the details observed in the quantum interference experiments, the above part has to be convincingly addressed.

- 5) The authors claim that magneto-conductivity is usually described by the HLN equation. Does

this also hold for the linear dispersion of surface states?

Reviewer #3 (Remarks to the Author):

The manuscripts reports on the temperature dependence of the dephasing rate for surface states of 3D topological insulators.

This dependence, extracted from the magnetoresistance measurements, shows a surprising gate-controlled crossover from the conventional linear-in-temperature behavior known for 2D diffusive systems to a sublinear behavior.

The authors attribute the enhancement of the dephasing to the inelastic processes involving the (almost) localized states in the 3D bulk. This is a very interesting and important study that will definitely contribute to our understanding of coherent processes and, more generally, of transport phenomena in topological insulators. The experimental results are very reliable, the proposed interpretation of the results is quite reasonable, and the conclusions are sound.

I recommend the publication after the authors have addressed the following two points:

1. It would be interesting to see the dependence of the extracted dephasing rate on the conductance of the 2D surface states. The point is that in similar experiments by Minkov's group on 2D topological insulators in the metallic regime (Fermi energy above the 2D band gap), this dependence was found to deviate from the conventional one. Of course, the emphasis in the present manuscript is put on the temperature dependence of dephasing, but showing the conductance dependence would be useful for interpretation of results.

2. It has been recently established that in a 3D system of localized states that interact via Coulomb interaction, the energy transport at low temperatures is much more efficient than the charge transport (power-law suppression vs. exponential suppression), see Gutman et al., Phys. Rev. B 93, 245427 (2016).

The Coulomb interaction of the 2D surface states with such "energy bath" can potentially be an efficient source of dephasing.

Reviewer #4 (Remarks to the Author):

The measurement of quantum corrections to diffusive transport in 3D topological insulators has been reported by many authors ever since the initial interest began in these materials some six years ago. Many of the initial papers focused on overly simplistic interpretation of weak anti localization in samples with significant bulk conduction. Subsequent theory papers (e.g. Garate & Glazman) coupled with experimental advances in the ability to electrically gate the samples led to better analysis. Nonetheless, the literature is still full of papers that are incomplete in the thorough understanding of quantum corrections because one needs a wide enough range of temperatures and wide enough tuning range for the chemical potential.

At first, when I looked at the content of this manuscript, my reaction was to group this work into the same category and I was ready to advise that it be sent to a journal such as Physical Review B where specialists can delve into the details of the results. However, the paper surprised me: it reveals new physical insights into quantum diffusive transport in the 3D topological insulators that has been completely missed in the earlier literature. The authors are the first to make a really thorough analysis of quantum diffusive transport in a 3D topological film as the chemical potential is varied to transition between the coupled bulk-surface transport regime to one where the surface dominates. Their key result is very interesting: they show that in the coupled bulk-surface transport regime, the linear temperature dependence of the dephasing field (deduced from weak anti localization fits) is completely consistent with Nyquist dephasing for a 2D system. However,

when the chemical potential is tuned into the bulk band gap, the temperature becomes sub linear and this can be interpreted as arising from dephasing due to a coupling between the surface states and localized electron-hole puddles in the bulk states. This is not completely new physics since STM has shown the existence of these puddles. But it is nonetheless an elegant result since it is I believe the first credible evidence from electrical transport for the existence of electron-hole puddles in the bulk of 3D topological insulators when the chemical potential is gated into the bulk gap. The implications are quite profound and I believe that they will change the way we think about these materials.

The paper is very clearly written. I do not see any need for any changes, but I do have one minor suggestion: some readers not familiar with diffusive quantum transport may be confused by the fact that the coupled bulk-surface regime acts like a 2D system and not like a 3D system. The authors should mention somewhere that the dephasing length is much longer than the film thickness (it would be worthwhile to also mention the value of the deduced dephasing length). Finally, I would also suggest including an additional reference to relevant work that measured the temperature dependence of the dephasing length in 3D topological insulators through universal conductance fluctuations in mesoscopic topological insulator channels (NanoLetters 13, 2471 (2013)).

This a high quality paper and will be influential. I recommend publication in Nature Communications.

Reviewer #1:

Comments: In the manuscript, the authors studied the dephasing mechanism in topological insulators (TIs). By tuning the Fermi level of TIs with top and back gates, the bulk became insulating. In such decoupled surface transport regime, a sublinear power law dependence of dephasing rate is observed. The authors ascribed it to the existence of charge puddles in the bulk of TIs due to the small band gap and the compensation doping. This study not only reveals a possible new dephasing mechanism in TIs, but also gains insight into the ground state of TIs. It is important to the community of TIs and suitable for publication in Nature Communications. But if such bulk-state charge puddles do exist in the bulk of TIs, they are also expected to appear at the surface, as well as the surface-state puddles with different charge types. Can such bulk-state and surface-state puddles at the surface also lead to the dephasing of carriers? Can we observe any experimental phenomena related with such charge puddles in the decoupled surface transport regime?

Reply: We thank the reviewer for the valuable comments. If the Fermi level is tuned close to the Dirac point, the charge puddles of Dirac fermions can indeed exist on TI surfaces due to spatially varying electrostatic potential. Evidence for such puddles has been obtained from an STM experiment [Ref. 44, Beidenkopf *et al.*, Nat. Phys **7**, 939 (2011)]. The transport in this regime is, however, *qualitatively* different from *strongly* localized charge puddles in the bulk due to the Dirac nature of the surface states. In this work, the maximum resistivity for each surface is much less than h/e^2 (~ 26 k Ω) for the entire range of gate voltages, suggesting the transport is mainly diffusive in nature. In our dual-gated devices, the transport can be varied continuously from a regime of surface charge puddles (See Fig. 1d, the points near the resistance maximum, which is only about 4 k Ω) to one without surface charge puddles (also in Fig. 1d, at gate voltages larger than those near the resistance maximum). In both cases, the bulk states are in the variable hopping regime and the phase coherent transport on the top and bottom surfaces are decoupled. Interestingly, both cases seem to exhibit similar sublinear temperature dependence of the dephasing rate, implying a dominant role of the charge puddles in the bulk. Nevertheless, it remains an interesting open problem whether the surface puddles can lead to an observable effect on the electron dephasing in TI samples with much weaker disorder, or in other words, in the samples the conductivity of each surface can be tuned close to e^2/h . This requires much higher sample quality and may be addressed in our future work. In the revised manuscript, we have modified a sentence on the previous experiments on the charge puddles to make the readers aware of the difference between the surface and the bulk charge puddles. Please see the list of changes for details.

Reviewer #2:

Comments: Liao and co-workers report on enhanced electron dephasing in the three-dimensional topological insulator (Bi,Sb)₂Te₃. The authors tune their transport properties using the gating technique to probe different regimes in this material. In their comprehensive experimental study, the authors claim that they observe a change in the dephasing rate and attribute their observation to surface transport or coupling between surface states and

localized charge puddles in the bulk.

As mentioned by the authors in the introductory part, literature is full of electron dephasing experiments on topological insulators and I do not see that the present paper has the novelty to be of interest for a broad physics community and therefore, I do not recommend publication of this paper in Nature Communications. This work could be interesting for a very limited community working on quantum interference experiments on novel materials and I think that Physical Review B might be an appropriate journal for publication.

Reply: We strongly disagree with the reviewer regarding the importance and novelty of the work. They have been described clearly in the original cover letter, as well as in the introduction and last few paragraphs of the manuscript. As pointed out by the other three referees, our work leads to a new dephasing mechanism that has not been discovered by the community despite a tremendous amount of previous work on this subject. Reviewer #4 further states that the implications (of our work) are quite *profound* and they will change the way we think about these (topological insulator) materials.

I have the following comments that should be addressed by the authors:

The authors claim that their dephasing experiments can disentangle different transport regimes in TIs, however, to my opinion, this assumption is not justified.

1) I do not see experimental evidence for surface-dominant transport. The authors use a two-carrier model and claim that the Fermi level is in the bulk band gap. What is the evidence for 2D transport? Can they exclude that a low-concentration bulk band instead of a surface state contributes to transport?

Reply: It seems that the reviewer ignored a large amount of knowledge accumulated in the past 7-8 years by the TI community on how to achieve the surface dominant transport and how to detect them. Many previous experiments on this subject have been cited in the introductory section of the manuscript. In the present work, we have not relied on the semiclassical two band fits alone to analyze the transport type, and a lot of effort has been made on using the weak antilocalization effect to probe the coupling between the surface and the bulk (See Fig.2 and the supplementary information). As we demonstrated previously, similar gate-tuning of ultrathin (Bi,Sb)₂Te₃ thin film leads to observation of Anderson localization [see Ref. 46, Liao et al., Phys. Rev. Lett. **114**, 216601 (2015).] Had the low-concentration bulk band existed, such strong localization behavior would have been impossible. Furthermore, similar gate-tuning of magnetically doped (Bi,Sb)₂Te₃ thin films grown in the same MBE machine resulted in the first observation of the quantum anomalous Hall effect [Ref. 12, Change et al., Science **340**, 167 (2013)], which also required the bulk conductivity to be suppressed.

2) How should I understand the statement “the surfaces are fully decoupled” or “gradual decoupling between top and bottom surface”? Do the authors mean hybridization? Is there any experimental evidence for that?

Reply: Here “the fully decoupled surfaces” means that the HLN prefactor $\alpha=1$ is observed. This requires that both the bulk conductivity and the coupling via the bulk states can be neglected.

Throughout this work, the film thicknesses are too thick to have substantial hybridization effect. The gradual decoupling means that the variation of the gate voltage gradually reduces the bulk conductivity as well as the coupling strength between the top and bottom surfaces. Such physics has been explained extensively in our previous work and by many other groups (See for instance, Refs. 11, 15-17, 38).

3) How does the total concentration depend on the applied gate voltage(s)?

Reply: The lower part of Fig.1d in the main text shows the gate-voltage dependence of the Hall coefficient R_H . The total carrier concentration can be estimated with $R_H=1/ne$, if the transport is not in the ambipolar regime. The total carrier density varies monotonically with gate voltages due to the electrostatic effect.

4) Can the authors comment on the presence of trivial surface states and possible geometric effects that can be present in thin TI films?

Reply: Topologically trivial surface states have been observed on the top surfaces of TIs by some ARPES experiments [see, e.g., M. Bianchi *et al.*, Nat. Commun. **1**, 128 (2010), P. D. C. King *et al.*, Phys. Rev. Lett. **107**, 096802 (2011)]. They are, however, irrelevant to our present work, because we used dual-gating devices. For the symmetric decoupled surface transport, the carrier densities are usually less than 10^{-12} cm⁻² on the top surface, which is too low to coexist with the trivial surface states formed due to band bending near the top surface. A statement in the Supplementary Information has been added in the revised manuscript. Please see below for the list of changes.

Before attributing the different regimes to the details observed in the quantum interference experiments, the above part has to be convincingly addressed.

5) The authors claim that magneto-conductivity is usually described by the HLN equation. Does this also hold for the linear dispersion of surface states?

Reply: The magnetoconductivity of 2D Dirac electron systems can be described by the HLN equation. Please see Suzuura & Ando, Phys. Rev. Lett. 89, 266603 (2002), Lu *et al.*, Phys. Rev. Lett. 107, 076801 (2011) and many other theoretical works on the weak antilocalization and localization effects in graphene and TIs.

Reviewer #3:

Comment: The manuscripts reports on the temperature dependence of the dephasing rate for surface states of 3D topological insulators. This dependence, extracted from the magnetoresistance measurements, shows a surprising gate-controlled crossover from the conventional linear-in-temperature behavior known for 2D diffusive systems to a sublinear behavior. The authors attribute the enhancement of the dephasing to the inelastic processes involving the (almost) localized states in the 3D bulk. This is a very interesting and important

study that will definitely contribute to our understanding of coherent processes and, more generally, of transport phenomena in topological insulators. The experimental results are very reliable, the proposed interpretation of the results is quite reasonable, and the conclusions are sound.

Reply: We thank the reviewer for a very careful review and positive remarks on our work.

I recommend the publication after the authors have addressed the following two points:

1. It would be interesting to see the dependence of the extracted dephasing rate on the conductance of the 2D surface states. The point is that in similar experiments by Minkov's group on 2D topological insulators in the metallic regime (Fermi energy above the 2D band gap), this dependence was found to deviate from the conventional one. Of course, the emphasis in the present manuscript is put on the temperature dependence of dephasing, but showing the conductance dependence would be useful for interpretation of results.

Reply: We thank the referee for pointing out an interesting work on a 2D TI. In 3D TIs, experimental study of the dephasing rate on the conductance is much more challenging. Such analysis can be done with help of the two-channel model in the decoupled surface transport regime. The parameter range is, however, too narrow to make a reliable conclusion. For instance, the sheet resistance of Sample #1 (the 15 nm BST sample presented in the main text), can only be varied from 3 k Ω to 4.2 k Ω in this regime (upper part of Fig. 1d in the main text). Further increasing in the conductivity makes the transport no longer in the decoupled regime, in which three conduction channels, including the top and bottom surfaces, and the bulk, contribute to the transport. It then becomes impractical to extract the conductivities and dephasing rates for all three channels reliably (See discussions in Sec 2.2 of the Supplemental Information). Therefore, we choose not to directly present the dependence of dephasing rate on the conductivity in this manuscript. Nevertheless, it would be very interesting if such analysis can be done by further increasing the quality of 3D TIs, in which a larger range of tuning in the conductivity can be achieved. In order to motivate potential readers, the paper of Minkov et al. has been added as Ref. 51 in the revised manuscript.

2. It has been recently established that in a 3D system of localized states that interact via Coulomb interaction, the energy transport at low temperatures is much more efficient than the charge transport (power-law suppression vs. exponential suppression), see Gutman et al., Phys. Rev. B 93, 245427 (2016). The Coulomb interaction of the 2D surface states with such "energy bath" can potentially be an efficient source of dephasing.

Reply: We thank the reviewer for the valuable and insightful suggestion. In the revised manuscript, we have added the following statements: *"It is also interesting to note that a recent theoretical work showed that the energy transport in the VRH regime is much more efficient than the charge transport due to Coulomb interaction between the localized states⁵². It is likely that the Coulomb interaction of the TI surface states with the charge puddles in the bulk can potentially be an additional source of dephasing."* The work of Gutman et al. has been cited as Ref. 52.

Reviewer #4:

Comment: The measurement of quantum corrections to diffusive transport in 3D topological insulators has been reported by many authors ever since the initial interest began in these materials some six years ago. Many of the initial papers focused on overly simplistic interpretation of weak anti localization in samples with significant bulk conduction. Subsequent theory papers (e.g. Garate & Glazman) coupled with experimental advances in the ability to electrically gate the samples led to better analysis. Nonetheless, the literature is still full of papers that are incomplete in the thorough understanding of quantum corrections because one needs a wide enough range of temperatures and wide enough tuning range for the chemical potential.

At first, when I looked at the content of this manuscript, my reaction was to group this work into the same category and I was ready to advise that it be sent to a journal such as Physical Review B where specialists can delve into the details of the results. However, the paper surprised me: it reveals new physical insights into quantum diffusive transport in the 3D topological insulators that has been completely missed in the earlier literature. The authors are the first to make a really thorough analysis of quantum diffusive transport in a 3D topological film as the chemical potential is varied to transition between the coupled bulk-surface transport regime to one where the surface dominates. Their key result is very interesting: they show that in the coupled bulk-surface transport regime, the linear temperature dependence of the dephasing field (deduced from weak anti localization fits) is completely consistent with Nyquist dephasing for a 2D system. However, when the chemical potential is tuned into the bulk band gap, the temperature becomes sub linear and this can be interpreted as arising from dephasing due to a coupling between the surface states and localized electron-hole puddles in the bulk states. This is not completely new physics since STM has shown the existence of these puddles. But it is nonetheless an elegant result since it is I believe the first credible evidence from electrical transport for the existence of electron-hole puddles in the bulk of 3D topological insulators when the chemical potential is gated into the bulk gap. The implications are quite profound and I believe that they will change the way we think about these materials.

The paper is very clearly written. I do not see any need for any changes, but I do have one minor suggestion: some readers not familiar with diffusive quantum transport may be confused by the fact that the coupled bulk-surface regime acts like a 2D system and not like a 3D system. The authors should mention somewhere that the dephasing length is much longer than the film thickness (it would be worthwhile to also mention the value of the deduced dephasing length). Finally, I would also suggest including an additional reference to relevant work that measured the temperature dependence of the dephasing length in 3D topological insulators through universal conductance fluctuations in mesoscopic topological insulator channels (NanoLetters 13, 2471 (2013)).

Reply: We thank the reviewer for a very careful review and for the valuable comments. We have made the following changes in the manuscript, following the suggestions of the reviewer.

- (1) A statement on the dephasing length vs film thickness has been added on page 5.

(2) The paper of Kandala et al. has been added as Ref. 20.

REVIEWERS' COMMENTS:

Reviewer #1 (Remarks to the Author):

The authors have addressed my concerns and clarified the role of surface puddles in the dephasing. Now the manuscript is recommended for publication in Nature Communications.

Reviewer #3 (Remarks to the Author):

I am satisfied with the authors' response to my comments, as well as with authors' responses to the comments of other referees and the changes made to the manuscript. The importance of this work is perfectly confirmed by the reports of reviewers 1 and 4. I find the rebuttal of the criticism of reviewer 3 by the authors fully convincing. In my view, the manuscript can now be accepted for publication.

A minor point:

After the first round of refereeing, I became aware of recent works by the Indian group who investigated the weak antilocalization in similar 3D topological insulators. They also observed a peculiar T dependence of dephasing and attributed it to the structural inhomogeneity of the system (in particular, to the granularity of the system), see arXiv preprint arXiv:1604.03767, arXiv preprint arXiv:1605.01613, Journal of Physics: Condensed Matter 29 (18), 185001 (2017). The authors of the present manuscript might find it useful to comment on these works and to present arguments that would explicitly rule out the possibility of structural inhomogeneity of the bulk.

REVIEWERS' COMMENTS:

Reviewer #1 (Remarks to the Author):

The authors have addressed my concerns and clarified the role of surface puddles in the dephasing. Now the manuscript is recommended for publication in Nature Communications.

Reviewer #3 (Remarks to the Author):

I am satisfied with the authors' response to my comments, as well as with authors' responses to the comments of other referees and the changes made to the manuscript. The importance of this work is perfectly confirmed by the reports of reviewers 1 and 4. I find the rebuttal of the criticism of reviewer 3 by the authors fully convincing. In my view, the manuscript can now be accepted for publication.

A minor point:

After the first round of refereeing, I became aware of recent works by the Indian group who investigated the weak antilocalization in similar 3D topological insulators. They also observed a peculiar T dependence of dephasing and attributed it to the structural inhomogeneity of the system (in particular, to the granularity of the system), see arXiv preprint arXiv:1604.03767, arXiv preprint arXiv:1605.01613, Journal of Physics: Condensed Matter 29 (18), 185001 (2017). The authors of the present manuscript might find it useful to comment on these works and to present arguments that would explicitly rule out the possibility of structural inhomogeneity of the bulk.

Reply:

We thank all reviewers for their efforts and support to publish the manuscript in Nature Communications. In the following, we address the additional comment of reviewer #3.

We have read the three papers listed above very carefully. In this series of work, Kumar *et al.* studied Bi₂Se₃ thin films with large inhomogeneity, and they reported that the temperature dependence of electron dephasing rate could vary from linear to nearly quadratic as the morphology of the Bi₂Se₃ films changes. Such *superlinear* behavior is *opposite* to what we observed in our (Bi,Sb)₂Te₃ thin films in the topological transport regime, in which the temperature dependence of the dephasing rate follows a *sublinear* power law. Since our samples, the single crystalline thin films carefully prepared with molecular beam epitaxy (MBE), are very smooth and have large, atomically flat terraces, the physics related to the large inhomogeneity encountered by Kumar *et al.* is not relevant to our work. Nevertheless, we have added a statement on the surface smoothness of our samples in the Methods section and cited two papers showing the STM images of topological insulator thin films grown in the same MBE machine as the one used in this work.